# A Decade of CRISPR-Cas Gnome Editing in *C. elegans*

**DOI:** 10.3390/ijms232415863

**Published:** 2022-12-14

**Authors:** Hyun-Min Kim, Yebin Hong, Jiani Chen

**Affiliations:** Division of Natural and Applied Sciences, Duke Kunshan University, Kunshan 215316, China

**Keywords:** CRISPR, Cas, genome editing, *C. elegans*, genome engineering

## Abstract

CRISPR-Cas allows us to introduce desired genome editing, including mutations, epitopes, and deletions, with unprecedented efficiency. The development of CRISPR-Cas has progressed to such an extent that it is now applicable in various fields, with the help of model organisms. *C. elegans* is one of the pioneering animals in which numerous CRISPR-Cas strategies have been rapidly established over the past decade. Ironically, the emergence of numerous methods makes the choice of the correct method difficult. Choosing an appropriate selection or screening approach is the first step in planning a genome modification. This report summarizes the key features and applications of CRISPR-Cas methods using *C. elegans*, illustrating key strategies. Our overview of significant advances in CRISPR-Cas will help readers understand the current advances in genome editing and navigate various methods of CRISPR-Cas genome editing.

## 1. Introduction

CRISPR-Cas9 (clustered regularly interspaced short palindromic repeats-associated) is the current choice for genome editing. It is an RNA-guided system where a crRNA and a trRNA, together called a guide RNA (sgRNA), direct a Cas9 nuclease to the target gene of interest (Figure 1, [1]). The crRNA consists of a 20-nucleotide sequence from the spacer of the CRISPR locus and corresponds to a target DNA. The trRNA is complementary to a pre-crRNA, thus forming an RNA duplex later cleaved by RNase III to form a crRNA–trRNA hybrid, thereby directing the Cas9 RGN to make a double-stranded break (DSB) at the target site. The DSB can then be repaired by non-homologous end joining (NHEJ) or homologous recombination (HR)/homology-directed repair (HDR). In *C. elegans*, CRISPR-Cas9 technology was first adopted in 2013 (Figure 2), and since then, the nematode-research community has produced increasingly sophisticated strategies for genome editing. Here, we introduce the significant discoveries of CRISPR-Cas genome editing achieved in the last decade (Figure 2).

## 2. History of *C. elegans* CRISPR-Cas

Since the first publication of the effects of the CRISPR-Cas9 nuclease in vitro in 2012 [3], parallel studies in different organisms have proven the role of molecular scissors approaches in vivo. The first demonstration of CRISPR-Cas9 in *C. elegans* involved the simple generation of loss-of-function mutants via a non-homologous end-joining (NHEJ) pathway [6]. This study targeted *unc-119* and *dpy-13,* which enabled screening of the transgenic mutants by screening uncoordinated (unc) or dumpy (dpy) phenotypes. Additionally, it demonstrated that a gene (*klp-12, Y61A9LA.1*) exhibiting no distinct phenotypes can also be identified by PCR screening methods. Single guide RNA (sgRNA), which combines crRNA and trRNA, can cleave target sites, thus eliminating the use of crRNA and trRNA separately, as demonstrated in *S. pyogenes* [3]. However, another report revealed that chimeric sgRNA is not as efficient as individual crRNA and trRNA [7].

Previously, homologous recombination in *C. elegans* was neither an efficient nor a major recombination pathway for double-stranded breaks [22,23]. However, this idea was no longer valid after the appearance of CRISPR-Cas. Soon after the first publication of NHEJ, multiple articles were released, with swiftly varying strategies. Three back-to-back reports demonstrated that CRISPR-Cas9 could edit the *C. elegans* genome via homologous recombination [7,8,9]. Hence, CRISPR-Cas-mediated homologous recombination (HR) allows a promising path for customized precise modifications.

## 3. Screening of Transgenic Worms

### 3.1. PCR and Drug-Based Selection

Whereas CRISPR induces DNA double-strand break (DSB) at the target genome, and the cell’s DSB repair system completes genome editing, the screening of the corrected genome still remains for scientists to complete. PCR has been the breakthrough technique for screening targeted mutations, and it is still helpful for most small-scale screenings [1]. However, for a large-scale genetic screening approach, it is often not time- and cost-efficient. Drug selection provides a new path, in addition to phenotype-based screening, to efficiently identify homologous recombination. Chen et al. incorporated the hygromycin resistance gene at the *ben-1* locus, endowing both hygromycin and benomyl resistance [8]. *Dpy-3* was also proposed as a new CRISPR-Cas tool, since its homozygous and hemizygous phenotypes are distinct [10].

### 3.2. Co-CRISPR and Co-Conversion: A Mutation in an Endogenous Marker Presents A Visible Phenotype

In 2014, *C. elegans* genome editing entered a new phase by adopting a second endogenous marker. Kim et al. introduced the Co-CRISPR strategy to detect genome editing. In addition to the gene of interest, *unc-22* was targeted and selected by identifying twitching worms. The *unc-22* indicates active Cas9 expression and a twitcher-based indel frequency via NHEJ mutations (Figure 3, [11]). Similarly, Arribere et al. proposed targeting a gene of interest by hitting second markers, such as *dpy-10* and *sqt-1* [12]. Since this strategy relies on homologous recombination events, it was named co-conversion. Co-CRISPR and co-conversion are similar in the sense that both methods target the gene of interest and an additional marker gene that confers phenotypic changes, thus ensuring that genome editing is being processed. Co-CRISPR uses NHEJ for targeting marker genes, whereas co-conversion relies on homologous recombination (HR) via template DNA injected into the germline.

### 3.3. Streamlined Genome Editing

In 2015, another set of new strategies was introduced to minimize hands-on labor and screening procedures. The streamlined methods incorporated three key features: a drug-resistance gene, a fluorescence marker, and Cre-Lox recombinase technology for seamless genome editing (Figure 4). The essential advantage of these methods is that minimal PCR screening is required to identify recombination by employing antibiotic resistance and a fluorescence reporter.

Notably, Norris and Kim et al. reported strategies to find recombinant versus extrachromosomal arrays by observing the mosaic expression of GFP and mCherry markers [13]. Studies from two independent labs described seamless genome editing by removing scars with Cre recombinase [13,14]. While both methods required Cre recombinase expression for seamless editing, Dickinson et al. eliminated the second microinjection of Cre by incorporating inducible Cre recombinase in the repair template [14].

## 4. Advances in Genome Editing Components

### 4.1. Cas9 Variants

Over the past decade, several CRISPR nuclease variants have been developed, and the range of its targets has been expanded (Figure 5). Nickases create a single-strand rather than a double-strand break. Targeting with nickase and two adjacent sgRNAs creates double-strand breaks with overhang, with reduces the off-target effects compared to the canonical/wild-type Cas9 system [4]. Dead Cas9 (dCas9) is an inactive catalytic nuclease. While the native Cas9 induces double-strand breaks on target DNA [5], dCas9 provides a binding site for activators or enhancers, promoting gene expression.

Since the *C. elegans* genome is an AT-rich genome, targeting conventional NGG near DSBs is frequently challenging. Therefore, having alternative sets of PAM sequences would expand the regions capable of being genome-edited. Kleinstiver et al. reported that Cas9 VQR (which possesses the amino acid substitutions D1135V, R1335Q, and T1337R) recognizes NGA PAM sequences instead of canonical NGG [24]. VQR recognizes NGAG as efficiently as wild-type Cas9 targets NGG [15], thus broadening the range of targets. Additionally, SpG and SpRY are two modified versions of Cas9, with more relaxed PAM requirements than Cas9 [25]. Thus, they target more portions of the genome. Of note, SpG and SpRY performed as efficiently as wild-type Cas9 at an increased concentration of CRISPR-Cas reagents (8 µM in the injection mix).

### 4.2. Guide RNA

Since guide RNA must designate the unique site of a gene of interest, multiple factors must be considered to achieve a high efficacy and specificity regarding genome editing. Studies have investigated strategies for designing guide RNA to enhance the genome editing frequency. For example, Farboud et al. reported that a GG motif at the 3′ end of the target sequences induced a high frequency of mutagenesis via NHEJ and HR pathways [16]. The median frequency at the targets was 10-fold higher than those reported in previous studies with multiple guide RNAs [6,11,26].

Later, the same group explored the location of DSB repair events in CRISPR-Cas editing. Interestingly, NHEJ induces asymmetric insertions and deletions (indels) preferentially in regions of 5′ of PAM [27]. Notably, a similar propensity for repair has been found in mammals, advocating the use of *C. elegans* as a model to understand the universal rules underlying genome editing [28,29].

Since variants of Cas nucleases have expanded the range of genome targets, identifying a proper target may now involve additional effort. Currently, a web-based database of guide RNA can help to identify guide RNAs for target genes and minimize off-target sites. Several database services can identify potential guide RNA for the entire *C. elegans* genome (genome.sfu.ca/gexplore (accessed on 4 December 2022), www.crisprscan.org (accessed on 4 December 2022), crispr.dbcls.jp (accessed on 4 December 2022), crispor.tefor.net (accessed on 4 December 2022) [30,31,32,33]). The user can search the guides present in the database by entering multiple factors, including genomic interval, GC content, gene name, and the presence of GG at the 3′ end of the guides. Notably, CRISPRscan can identify targets for the Cas9 variants SpG and SpRY.

### 4.3. Dual sgRNA/Dual DSBs

While Cas9 nuclease induces a single DSB at a target site, dual or triple DSBs offer advantages for genome editing. For example, deletion mutants can be achieved by adopting two sgRNAs via NHEJ or HR pathways (Figure 6A, [17]). Likewise, dual sgRNA can generate long chromosome deletion between two sgRNAs [18]. Chen et al. also demonstrated that dual DSBs generate reciprocal chromosomal translocation, thereby providing a practical approach to studying genome rearrangement [34].

Zhang et al. reported that dual cutting of the repair template, rather than genomic DNA, enhanced precise genome editing efficacy (Figure 6). The dual-cut repair template, flanked by two sgRNA at both ends of the plasmid, increased HR efficiency by between two and fivefold compared to a conventional uncut circular template. They observed a proportional increase in HR efficiency, with a more extended homologous arm in either circular or linear DNA, suggesting that a longer flanking sequence improves dual-cut-mediated HR efficacy [35]. Similarly, dual DSBs promoted the insertion of large (9300 bp) DNA fragments, in combination with a *dpy-10* co-conversion strategy, where single DSBs failed (5% vs. 0%, respectively) [27]. It is worth noting that, when the dsDNA template was cleaved by two DSBs, orienting both PAMs outward resulted in efficient homologous recombination (21% vs. 12% in an out/out and in/in configuration, respectively, Figure 6).

### 4.4. Oligonucleotide as a Repair Template for Homologous Recombination

Since linear DNA is prone to degradation by nucleases present in the cells, the conventional way of delivering a repair template is to supply double-stranded DNA as a part of circular DNA. Therefore, plasmid DNA is desirable, especially for large-sized repair templates. However, a growing number of studies have found that single-stranded oligonucleotides can often be good alternatives for HR-mediated genome editing. Specifically, oligonucleotides offer a few advantages over plasmid DNA: they are cloning-free and can be rapidly synthesized via commercially available resources. Therefore, adopting linear DNA reduces the amount of time required for the entire genome editing process.

The first demonstration of oligonucleotides as repair templates was performed in 2014, substantiating their simplicity and efficacy [19]. An oligonucleotide, ~100 bp long, serves as a template to repair DSBs occurring at four different genes via homologous recombination (Figure 7). This strategy has demonstrated its efficiency in several labs [12,36,37]. It is worth noting, however, that it appears that smaller flanking homology requires the DSB/cleavage site to be near the site of genome editing [12].

### 4.5. Modified Repair Template

Previous research investigated whether modification of the donor template improves HR efficacy. Although controversial at first, a growing number of studies have reported that the modification of repair donors enhances genome editing proficiency (Figure 7). In 2014, Zhao et al. employed a phosphorothioate-modified oligonucleotide as a repair template in *C. elegans* for the first time. Although the modified oligonucleotide resulted in the desired genome editing, it was unclear whether its efficacy was enhanced compared that resulting from to non-modification [19]. In mammalian cultured cells, however, phosphorothioate-modified oligonucleotides enhanced the genome editing efficiency of single-stranded oligonucleotide donors. In addition, modified oligonucleotides allow for insertions > 100 bp long, providing design flexibility [37]. In contrast, fluorescent and amine modifications to the 5′- and 3′-termini of single-stranded oligodeoxynucleotide (ssODN) donors did not alter HR frequency compared to nonmodified donors in human cells [38].

With this controversy, recent studies have explored donors with 5′-end modifications that enhance genome editing efficacy. Gutierrez-Triana et al. showed that adding biotin at the 5′ ends of dsDNA leads to an increase in HR efficiency of up to 60% in the injected generation of medaka fish embryos. Provocatively, the authors demonstrated that biotin and SpC3 5′ modifications prevent donor multimerization/NHEJ of dsDNA, thereby providing optimal conditions for HR-mediated CRISPR-Cas genome editing [39]. Similarly, Yu et al. reported that 5′ C6-PEG10-modified dsDNA increased knockin frequency up to fivefold, in combination with Cas9 ribonucleoprotein (RNP) in human cells [40]. In line with these reports in fish and mammalian cells, the *C. elegans* study demonstrated that 5′ modifications of the donor improved the efficacy of HR frequency roughly twofold [41]. Interestingly, TEG (triethylene glycol) and RNA::TEG modifications performed with similar efficacy in *C. elegans*, whereas RNA::TEG was superior to TEG in human cells and zebrafish. They demonstrated that these 5′ modifications suppress donor ligation reactions in a similar way to that of biotin and SpC3 5′ modifications [39].

It is worth noting that *C. elegans* studies from the Mello Lab reported additional alterations that enhance HR efficiency dramatically by modifying repair template DNA. First, a single-stranded overhang containing dsDNA donors yielded higher integration rates at three loci [36]. Second, denaturing and cooling the dsDNA donor template increased the HR frequency by up to 50% [20].

## 5. Other Developments

### 5.1. Off-Target Effects

Given that genome editing relies on the creation of DSB, potential off-target effects have been one of the major concerns. The off-target effects of CRISPR-Cas9 have been addressed a few times in *C. elegans* studies. Chiu et al. performed high-throughput sequencing after CRISPR-Cas genome editing and found no distinct evidence of off-target genomic lesions using GATK pipeline or split-read analysis [42]. Second, whole-genome sequencing of five strains revealed no distinct mutations at predicted off-target sites [17]. These suggest that off-target effects can be avoided by the accurate designing of guide RNA, with the help of computational tools.

### 5.2. CRISPR-Cas Genome Editing Protocols

We have listed some of the *C. elegans* genome editing protocols. Please see the list of major *C. elegans* CRISP-Cas publications for more information (Appendix A).

Efficient Genome Editing in *Caenorhabditis elegans* with a Toolkit of Dual-Marker Selection Cassettes [13].Precision genome editing using CRISPR-Cas9 and linear repair templates in *C. elegans* [43].An Efficient Genome Editing Strategy to Generate Putative Null Mutants in *Caenorhabditis elegans Using CRISPR/Cas9* [44].CRISPR-Cas9-Guided Genome Engineering in *Caenorhabditis elegans* [1].An affordable plasmid miniprep suitable for proficient microinjection in *Caenorhabditis elegans* [45].Design of Repair Templates for CRISPR-Cas9-Triggered Homologous Recombination in *Caenorhabditis elegans* [46].Microinjection for precision genome editing in *Caenorhabditis elegans* [47].High-efficiency CRISPR gene editing in *C. elegans* using Cas9 integrated into the genome [48].Approaches for CRISPR/Cas9 Genome Editing in *C. elegans* [49].CRISPR/Cas9 Methodology for the Generation of Knockout Deletions in *Caenorhabditis elegans* [50].CRISPR-Based Methods for *Caenorhabditis elegans* Genome Engineering [51].

## 6. Perspectives

### 6.1. Pros and Impact

CRISPR represents the greatest revolution in gene editing to date. With the advance of CRISPR-Cas genome editing technology, scientists have continued to expand this revolution for over a decade. The impact of precise and robust genome editing is broad and potent and has thus been able to influence the plant industry, scientific research, livestock improvement, and biomedical engineering, as well as human diseases and fertility. Furthermore, it has not just been limited to microbes, but has also been applied to mammalian cells, animal models, and plants, as well as for the modification of secondary metabolites, such as antibiotics [52], and medicinally bioactive compounds, including morphine and thebaine in opium. The unprecedented potential and impact of CRISPR-Cas have also produced concerns. Misuse of gene editing could trigger risks and dangers beyond imagination; therefore, urgent ethical concerns need to be addressed. In this regard, *C. elegans* is an ethical-issue-free animal model that can substitute for some portion of the conventional animal models used in research labs.

### 6.2. C. elegans for CRISPR-Cas Genome Editing

*C. elegans* has served as the leading animal model of CRISPR-Cas. The success of *C. elegans* as a CRISPR-Cas genome editing model lies in its large progeny (∼300 eggs), rapid life cycle (~3 days), self-fertilization, and easy maintenance [53,54]. Strategies established in *C. elegans* were successively adopted in other nematode species. The strategies work in *C. remanei*, *C. briggsae*, and the parasitic nematodes *Strongyloides spp.* and *A. freiburgensis* [55,56,57,58,59,60]. Specifically, CRISPR-Cas9 in *C. briggsae* can utilize *C. elegans* promoters to express sgRNAs and Cas9 and, similar to *C. elegans,* can use the 3′ GG motif to increase genome editing efficacy [61]. Thus, applying CRISPR-Cas discovered in *C. elegans* to other nematodes would provide a powerful tool to investigate the functions of orthologs, thereby leading to a better understanding of the molecular mechanisms. In contrast, there are intrinsic hurdles in *C. elegans* genome editing. Although some studies developed a quantity drug screening in *C. elegans*, a high-throughput CRISPR screening approach for identifying genes or pathways is not feasible. This is mainly because genome editing relies on a slow, multi-step microinjection process. Given that a recent study demonstrated the successful delivery of sgRNA [21], feeding-based CRISPR may be an alternative method for genome editing.

Studies over the last decade have demonstrated that improvements in genome editing can unlock the enormous potential that CRISPR offers. However, CRISPR-Cas is still far from perfect. Consequently, we are expecting various aspects of progress. For example, while conventional strategies rely on type II Cas9, the remaining five CRISPR systems have not been well investigated. Studies on the remaining types expand the targets of CRISPR-Cas application. For example, Cas12 of type V induced staggered breaks in dsDNA, unlike blunt breaks created by Cas9 [62]. Cas13 of type VI targets RNA, not DNA [63]. Second, improved delivery systems may enhance genome editing efficacy. Since CRISPR-Cas9 requires a large Cas9 complex (RNA-Protein) to cleave a target DNA in the nucleus, the low delivery efficiency restricts its successful editing. CRISPR-Cas combined with viruses, lipids, and synthetic molecules may enhance the delivery of CRISPR-Cas9 to the target DNA (Figure 6, [64,65]).

Advances in new screening methods, secondary endogenous markers, dual DSBs, modified repair templates, computational tools, and Cas variants have extended the efficiency and specificity of genome editing. This further strengthens overall genome editing strategies and incentivizes their use in science, medicine, pharmaceuticals, and industry. Furthermore, given the high level of conservation between mammals and *C. elegans*—60–80% of *C. elegans* genes are conserved between the two groups of organisms [66,67]—the mechanism of genome editing and the DNA repair pathway found in *C. elegans* may also be applicable to mammalian systems.

## Figures and Tables

**Figure 1 ijms-23-15863-f001:**
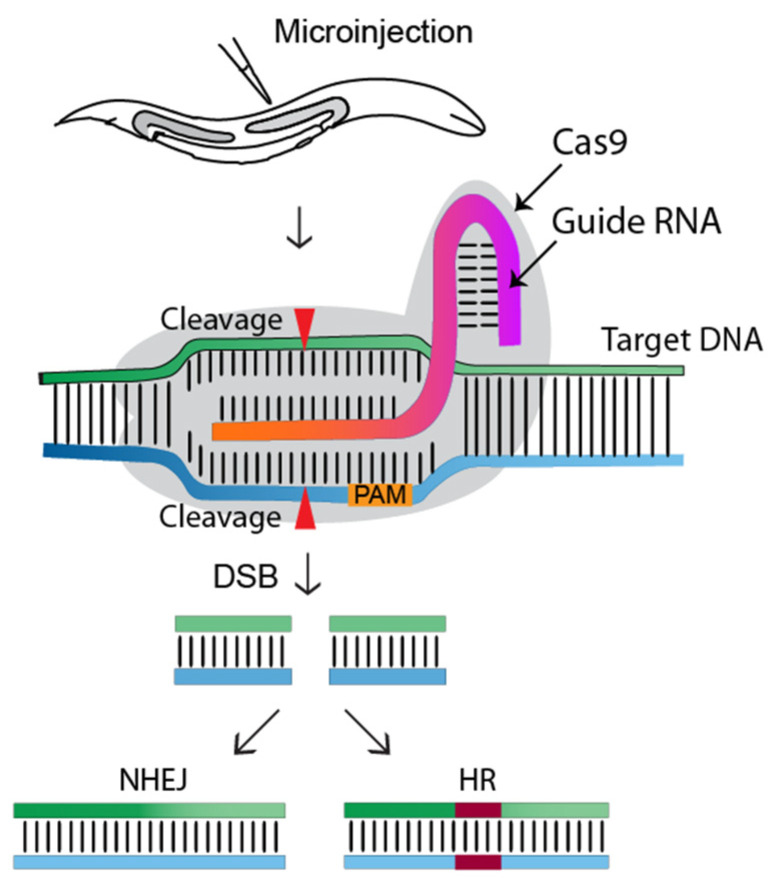
**Overview of the CRISPR-Cas9 genome editing approach in *C. elegans*.** The gonads of adult hermaphrodites are injected with the CRISPR-Cas9-containing DNA mixture. Injected Cas9 (grey color) forms a sequence-specific endonuclease when complexed with crRNA and trRNA (together called a guide RNA, or single guide RNA). The Cas9-guide RNA complex recognizes a target DNA sequence containing a PAM sequence (NGG) and induces a double-strand break (DSB), which will be repaired by non-homologous end joining (NHEJ) or homologous recombination (HR/HDR). Transgenic worms will be identified via marker-free (PCR) or marker-dependent strategies [1].

**Figure 2 ijms-23-15863-f002:**
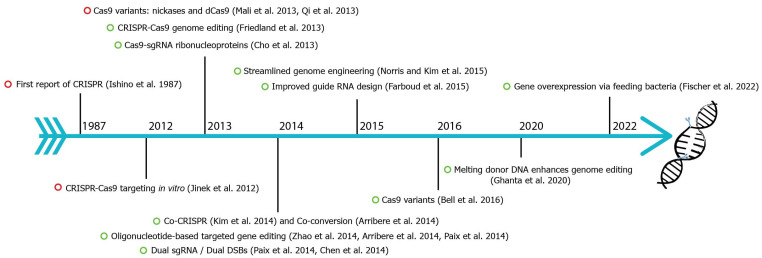
**Fundamental discoveries and advances in *C. elegans* CRISPR-Cas genome editing.** Timeline highlighting major events of *C. elegans* CRISPR-Cas9 (green circle) and other species (red circle). CRISPR were first identified in *E. coli* in 1987 [2,3,4,5,6,7,8,9,10,11,12,13,14,15,16,17,18,19,20,21].

**Figure 3 ijms-23-15863-f003:**
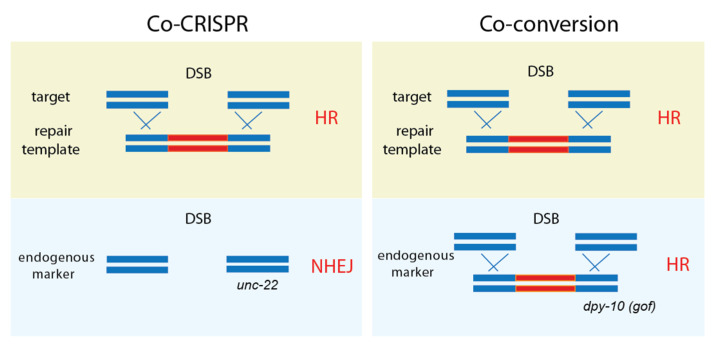
**Co-CRISPR and co-conversion.** Both methods employ an endogenous marker gene that exhibits phenotypic changes, thus ensuring that proper genome editing is processed. While Co-CRISPR uses NHEJ for targeting marker genes, co-conversion relies on homologous recombination (HR) via a repair template.

**Figure 4 ijms-23-15863-f004:**
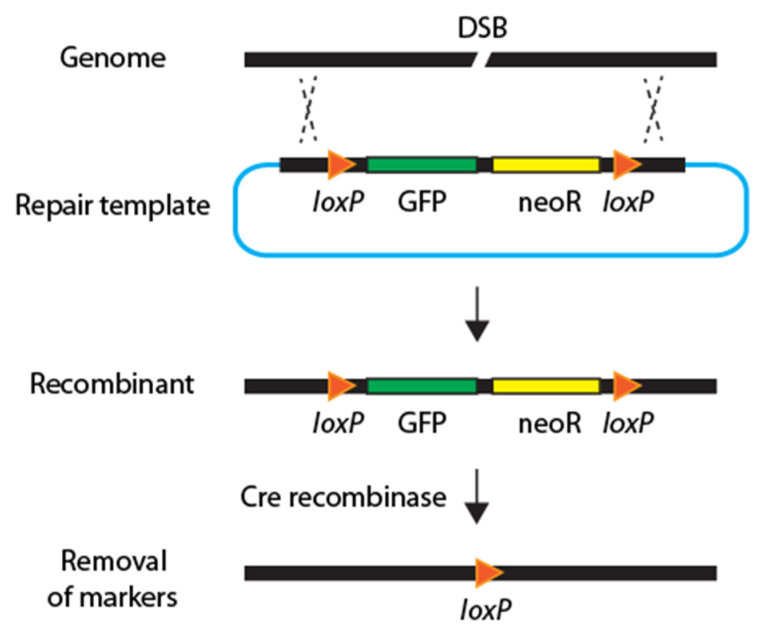
**Diagram of streamlined CRISPR-Cas**. The streamlined strategies integrate a drug-resistance gene (yellow), a fluorescence marker (green), and a Cre recombinase to facilitate smooth genome editing by removing these two markers.

**Figure 5 ijms-23-15863-f005:**
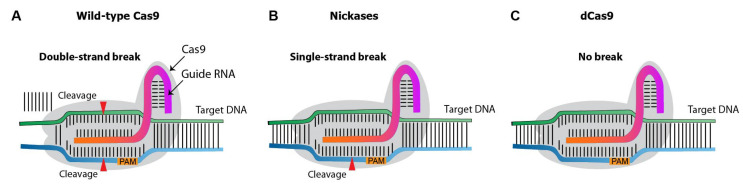
**Diagram illustrating different types of Cas9 variants**. (**A**), wild-type Cas9 nuclease. (**B**), Cas9 nickases that introduce a single-strand break at the target DNA. (**C**), catalytically inactive (dead) dCas9, which induces no breaks.

**Figure 6 ijms-23-15863-f006:**
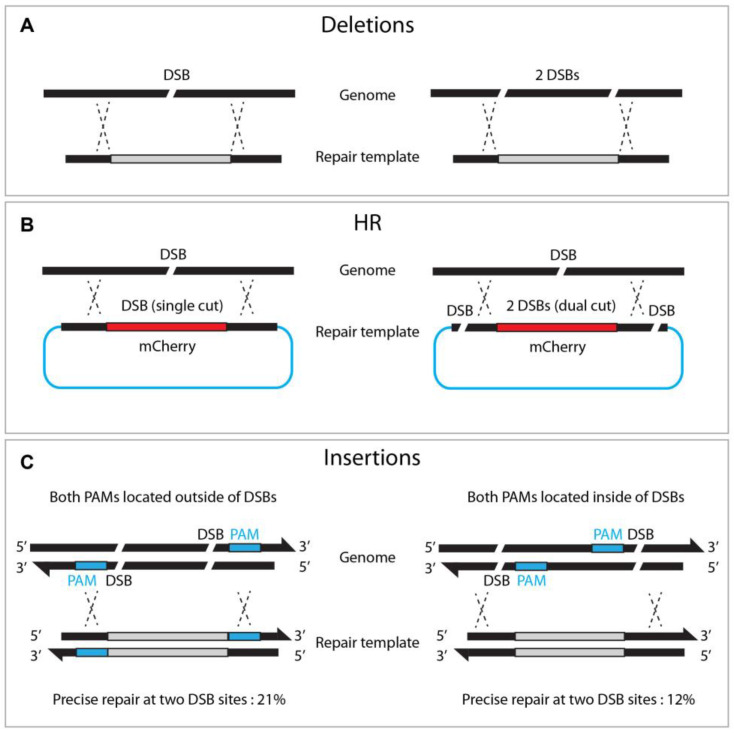
**Dual DSBs enhance deletions, homologous recombination, and insertions.** (**A**) Single versus dual DSBs at targeted sites. (**B**) Dual DSBs (right) in the repair template enhances HR efficacy compared to single DSB (left). (**C**) Locating both PAMs outward of the DSB sites enhances insertion frequency.

**Figure 7 ijms-23-15863-f007:**
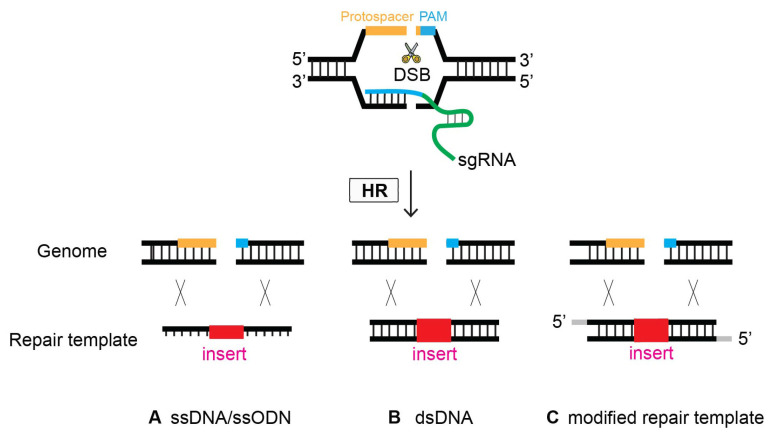
**Oligonucleotides can serve as a template to repair DSBs via HR.** At the site of CRISPR-Cas9-induced DSB, Single-stranded DNA (ssDNA), double-stranded DNA (dsDNA), and modified repair templates serve as a repair template. (**A**) ssDNA, (**B**) dsDNA, and (**C**) modified nucleotides as a repair template.

## Data Availability

Not applicable.

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
