# Peer review of "A Decade of CRISPR-Cas Gnome Editing in C. elegans"

_ijms, 2022, doi:10.3390/ijms232415863_

Round 1

Reviewer 1 Report

The article should be helpful to those interested in learning about the CRISPR-Cas genome editing in the C. elegans animal model. I am okay with the organization of contents but have the following suggestions.

There are many instances of spelling and grammar errors that are annoying. The following are a few examples, but it is not comprehensive list! Authors should review the entire text carefully to eliminate such errors.

Line 15: delete ‘to’
Lines 22 and 57: change "tRNA" to "trRNA"
Line 124: change "gRNA" to "sgRNA"
Line 159: change "ca" to "can"
Line 313: replace ‘is with ‘may’. One can’t be very certain about mechanisms being conserved.

Section 2.
I suggest getting rid of sub-sections. The paragraphs are rather small, so they will be fine without headings.

Section 4.2 (line 160)
Give some examples of databases, e.g., CRISPOR.

Concordet, J. P. and Haeussler, M. (2018). CRISPOR: Intuitive guide selection for CRISPR/Cas9 genome editing experiments and screens. Nucleic Acids Research, 46(W1), W242–W245. https://doi.org/10.1093/nar/gky354.

Section 6.
The sub-sections 6.1 and 6.2 are generic, not a good fit for the article, and are distracting. Please delete these. Alternatively, you may discuss the pro-cons and impact in one short paragraph but restrict contents to the C. elegans system.

Finally, please include the following reference in your article:
Dickinson, D. J. and Goldstein, B. (2016). CRISPR-based methods for Caenorhabditis elegans genome engineering. Genetics, 202(3), 885–901. https://doi.org/10.1534/genetics.115.182162.

Author Response

We greatly appreciate the reviewer’s comment. We agree with the reviewer’s assessment and incorporated changes in the text.

Reviewer 2 Report

The review article “A decade of CRISPR-Cas genome editing in C. elegans” by Kim et al. summarizes the significant advances in CRISPR-Cas methodology for model organism of C. elegans over the last decade. The authors first provide a brief overview of the history of the C. elegans CRISPR-Cas strategy, following the advantages of various strategies for screening of genome editing and justifying the pros and cons of other genome editing components such as Cas9 variants, sgRNA, and repair templates.

Despite the review is concise, it is well balanced and provides the readers with the current progress of the CRISPR-Cas strategies applied to genome editing of C. elegans.

In my opinion, the submitted review can be accepted after minor revision (see below).

Comments and Recommendations:

Line 22: The full meaning of crRNA and trRNA should be explained when first mentioned. tRNA should be corrected to trRNA. Check the manuscript for the missuses of tRNA instead of trRNA.

Line 51: The references for the use of CRISPR-Cas9 in other model organisms should be cited here.

Lines 160-161: Direct links to web-based tools and databases that allow researchers to identify potential guide RNA should be included and properly cited here.

Author Response

(The authors gave the same response as above.)

Reviewer 3 Report

In this review article, the authors provide a chronological overview of CRISPR development in C.elegans gene editing.

  • The authors should briefly outline the key steps of the CRISPR method in C. elegans. 
  • Authors should comment on the advantages of using C. elegans as a model to study CRISPR over other models.
  • Authors should include a table depicting the genome editing efficiencies of each CRISPR method and its modifications.
  • Subsection 6.2, Ethical concerns of genome editing, appears to be out of context. Authors should highlight the implication of C. elegans research to address ethical concerns of genome editing.
  • Authors should highlight the shortcomings of using C. elegans as a model to study CRISPR gene editing. 
  • Authors should highlight any seminal findings from C. elegans CRISPR studies that influenced later developments in nematode or other research models. 
  • It will be helpful if the authors provide a perspective on the future directions of CRISPR research in C. elegans. 

Minor: 

  • Line 29: please include the related reference.

Author Response

(The authors gave the same response as above.)

Round 2

Reviewer 3 Report

The authors have addressed almost all of my recommendations, and incorporating suggested changes has significantly improved the manuscript. 

However, an explanation for the shortcomings of using C. elegans as a model to study CRISPR gene editing needs to be more satisfactory. 

Line 306: This section should comment on the drawback of C. elegans' CRISPR technology: a high-throughput CRISPR screening approach for identifying genes or pathways is not feasible in C. elegans. Also, highlight the rate-limiting step of microinjection and the relatively cumbersome screening process to identify desired mutants. If known, authors can comment on how this drawback can be addressed in the future. The authors can omit the sentence about the lack of antibodies availability. 

Author Response

Second revision from the Reviewer #3

The authors have addressed almost all of my recommendations, and incorporating suggested changes has significantly improved the manuscript.

 However, an explanation for the shortcomings of using C. elegans as a model to study CRISPR gene editing needs to be more satisfactory.

Line 306: This section should comment on the drawback of C. elegans' CRISPR technology: a high-throughput CRISPR screening approach for identifying genes or pathways is not feasible in C. elegans. Also, highlight the rate-limiting step of microinjection and the relatively cumbersome screening process to identify desired mutants. If known, authors can comment on how this drawback can be addressed in the future. The authors can omit the sentence about the lack of antibodies availability.

We agree with the reviewer’s assessment and incorporated changes in the text and now it reads:

Line306. In contrast, there are intrinsic hurdles in C. elegans genome editing. Although some studies developed a quantity drug screening in C. elegans, a high-throughput CRISPR screening approach for identifying genes or pathways is not feasible. This is mainly because genome editing relies on a slow, multi-step microinjection process. Given that a recent study demonstrated the successful delivery of sgRNA [65], feeding-based CRISPR may be an alternative method for genome editing.